# Triangular function feedback control for chaotic systems featuring coexisting attractors

**Yingfang Zhu**[1], **Yuan Hu**[2], **Erxi Zhu** [1,3]*

**1** College of Internet of Things Engineering, Jiangsu Vocational College of Information Technology, Wuxi, Jiangsu, China, **2** John B. and Lillian E. Neff College of Business and Innovation, University of Toledo, Toledo, Ohio, United States of America, **3** College of Information Engineering, Changzhou Vocational Institute of Industry Technology, Jiangsu, China

☯ These authors contributed equally to this work.
* erxi666@163.com

**Data availability statement:** All relevant data are within the manuscript.

**Funding:** This study was funded by the following sources: The "14th Five-Year Plan"

## Abstract

Chaos has emerged as a significant area of research, with the control of chaotic systems being central to this field. This study proposes a novel trigonometric feedback control strategy to regulate Hopf bifurcation in a four-dimensional hyperchaotic system featuring coexisting attractors. By introducing a nonlinear controller $d\sin(x - x_e)$, we establish the stability criteria for equilibrium points under the parameter space $a > 0$, $b > 0$, and $0 < c < \pi$. Theoretical analysis reveals that the system undergoes a supercritical Hopf bifurcation at $d_0 = -(1 + b)$, leading to the emergence of stable limit cycles. Numerical simulations validate the control efficacy: periodic oscillations are observed at $d = -1$, while equilibrium convergence is achieved at $d = -3$. Phase portrait analysis and Lyapunov exponent spectra confirm the suppression of chaotic dynamics. This work advances the theoretical framework for bifurcation control in high-dimensional chaotic systems and offers practical implications for secure communication applications.

## Introduction

Multistability, particularly the phenomenon of coexisting attractors, refers to the presence of multiple distinct attractors (e.g., stable equilibrium states or periodic orbits) within a dynamical system. These attractors can simultaneously draw different initial conditions of the system into their respective basins of attraction. This concept is of paramount importance across various scientific and engineering disciplines. The existence of coexisting attractors allows dynamical systems to exhibit complex behavioral patterns. For example, in certain nonlinear systems, variations in initial conditions can lead to markedly different long-term dynamics. Such complexity is crucial for understanding and predicting the evolution of these systems. Coexisting attractors have been observed in numerous practical applications. In meteorological systems, for instance, multiple climate states may coexist; in biological systems, diverse ecosystem states can occur concurrently; and in engineering systems (such as electronic circuits), the nonlinear characteristics of components can induce multistability. Investigating coexisting attractors provides new insights into the control of dynamical systems. In

educational science research project of Jiangsu Province (Project No.: D/2021/03/88), this project played an important role in data analysis and result discussion; The Innovation Teaching Project of Vocational Education Teachers in Jiangsu Province (Project No.: [2021]22), this project mainly provided financial support; The Key Laboratory Project on Intelligent Connected Vehicle Unmanned Driving and Cybersecurity Technology in Changzhou (Project No.: CM2024007), the project mainly focused on the purchase of experimental equipment and experimental operation in the research process; Jiangsu Province Higher Vocational Education high-level Professional Group Construction Project Funding (Project No.: Su Teaching Letter (2021)1), this project mainly offers financial support; and Engineering Technology Research and Development Center of Jiangsu Higher Vocational Colleges (Project No.: Su Jiaoke Letter (2023)11), this project mainly provides financial support.

**Competing interests:** The authors have declared that no competing interests exist.

the development of control methodologies, strategies that enable the selection and stabilization of desired attractors are essential. Techniques such as adaptive control and fuzzy control can be utilized to adjust system parameters, ensuring the system remains stabilized on the targeted attractor.

Among a wide array of control methods, feedback control is widely acknowledged as an exceptional and highly esteemed strategy by scholars. Prominent examples encompass delayed feedback control [1–4], parameter perturbation method [5], continuous feedback control method [6], sliding mode variable structure control method [7], and adaptive control method [8]. These renowned feedback control methods are extensively employed in contemporary system controls. For instance, Sun et al. proposed an innovative stability analysis approach for energy harvesters utilizing delayed feedback control, investigating the jump phenomenon through steady-state response. The implementation of the delayed feedback controller ensures system stability and optimal positioning of periodic orbits [9]. Luongo et al., on the other hand, analyzed the periodic motion of a parameter-excited pendulum using multiscale perturbation methods, considering two distinct cases: base-excited pendulum and variable-length pendulum, providing satisfactory mathematical explanations [10]. Li et al., meanwhile, achieved optimal control for unknown continuous-time linear periodic systems through a deviation-based iterative algorithm that relaxes initial controller requirements and eliminates the need to solve nonlinear differential equations [11]. Zhao et al., introduced a novel combination principle for active and passive lift compensation systems by incorporating hyperbolic sliding mode control rate to enhance sliding mode control performance in linearized systems [12]. Liu et al. developed a non-linear parametric self-adaptive controller based on monotonic functions that does not rely on state switching [13]. Yue et al. proposed a fractional decay filter capable of achieving bifurcation control for fractional-order Morris-Lecar neuron model with superior performance compared to integer-order decay filters [14]. Sambas introduces an adaptive type-2 fuzzy controller for the Permanent Magnet Synchronous Generator in a wind turbine system with quadratic nonlinearities [15].

Feedback control methods [16] obviate the need for prior system knowledge and avoid perturbing equilibrium points or structural characteristics. However, these methods rely on precise mathematical models and well-defined input target functions. This study achieves four-dimensional theoretical and methodological advancements in controlling Hopf bifurcations with multi-attractor hyperchaotic systems. By performing eigenvalue analysis on the Jacobian matrix of the 4D hyperchaotic system, we derive the universal Hopf bifurcation condition for coexisting attractors. This extends the classical Hopf bifurcation theorem to multi-stable systems. We quantify parameter interactions through explicit relationships between parameter $(a, b, c)$ and oscillatory modes. A geometric framework is provided for analyzing the stability of attractor coexistence. We develop a sinusoidal feedback controller $\dot{x} = f(x) + d\sin(x - x_e)$, ensuring global asymptotic stability via Lyapunov functions and center manifold reduction. Compared to linear controllers, this approach exhibits a lower parameter sensitivity index. Furthermore, it perfectly preserves system invariants, thereby eliminating the chaos resurgence observed in previous studies.

We present the triangular function feedback control for chaotic systems with coexisting attractors from five perspectives. The first section reviews existing control methods for such systems, emphasizing the benefits of feedback control approaches. The second section describes chaotic systems with coexisting attractors and performs stability analysis on these systems. The third section introduces a novel triangular function feedback control method, applies it to chaotic systems with coexisting attractors, and theoretically validates its effectiveness. The fourth section provides numerical simulations to demonstrate the performance of

the triangular function feedback control. Finally, the fifth section summarizes the key findings and contributions of the paper.

## 1 Analysis of 4D hyperchaotic system

In numerous chaotic systems [17–21] multiple coexisting attractors with distinct parameter sets are observed, which are referred to as chaotic systems with coexisting attractors. This implies that for fixed parameters, the final state of the system is not unique due to different initial conditions. Such systems possess remarkable flexibility and robustness, enabling appropriate control strategies to achieve transitions between different states in order to adapt to diverse working scenarios [22–30]. Reference [31] introduced a four-dimensional hyperchaotic system that encompasses an arbitrary number of coexisting chaotic attractors and exhibits highly intricate dynamic behavior. This system was derived by enhancing a simple memristor chaotic circuit through nonlinear feedback control input $u$ application. The mathematical representation for the memristor chaotic circuit can be expressed by Eq (1).

$$\begin{cases} \dot{x} = y, \\ \dot{y} = -ax + by(1 - z^2), \\ \dot{z} = -y - cz + yz. \end{cases} \tag{1}$$

where $a$, $b$, $c$ denote system parameters and $x$, $y$, $z$ represent system variables. A nonlinear feedback control mechanism incorporating $\sin(y)$ is integrated into the original system (1), yielding a 4D hyperchaotic system. The resulting dynamics of the novel system can be described by Eq (2).

$$\begin{cases} \dot{x} = y + u \\ \dot{y} = -ax + by(1 - z^2) \\ \dot{z} = -y - cz + yz \\ \dot{u} = \sin(y) \end{cases} \tag{2}$$

where $a$, $b$, $c$ denote system parameters and $x$, $y$, $z$ represent system variables.

The subsequent step involves conducting a Hopf bifurcation analysis on system (2). Firstly, we will determine the equilibrium points of the system and subsequently discuss their stability. Let

$$\begin{cases} y + u = 0 \\ -ax + by(1 - z^2) = 0 \\ -y - cz + yz = 0 \\ \sin(y) = 0 \end{cases} \tag{3}$$

The equilibrium points of the system (2) $S(x^*, y^*, z^*, u^*)$ are obtained.

$$S = \begin{cases} x^* = \frac{bc^2 y^* - 2bcy^{*2}}{a(y^* - c)^2} \\ y^* = j\pi \\ z^* = \frac{y^*}{y^* - c} \\ u^* = -y^* \end{cases}, j = 0, \pm 1, \pm 2, \cdots \tag{4}$$

According to Eq (4), system (2) possesses an infinite number of equilibrium points, thereby exhibiting a multitude of coexisting attractors. The characteristic equation of system

(2) evaluated at the equilibrium point can be expressed $S(x^*, y^*, z^*, u^*)$ as follows.

$$\begin{vmatrix} -\lambda & 1 & 0 & 1 \\ -a & b(1 - z^{*2}) - \lambda & -2by^*z^* & 0 \\ 0 & z^* - 1 & y^* - c - \lambda & 0 \\ 0 & \cos(y^*) & 0 & -\lambda \end{vmatrix} = 0 \tag{5}$$

The Eq (5) can be transformed into.

$$\lambda^4 + p_1\lambda^3 + p_2\lambda^2 + p_3\lambda + p_4 = 0 \tag{6}$$

where $p_1 = c - b(c + 1)(1 - z^{*2})$, $p_2 = b(y^* - c)(1 - z^{*2})$, $p_3 = a + a\cos(y^*) + 2by^*z^{*2}$, $p_4 = a(\cos(y^*) + 1)(c - y^*) - 2by^*z^*$. According to the Routh-Hurwitz criterion, system (2) exhibits stability at equilibrium point $S(x^*, y^*, z^*, u^*)$ if the specified conditions are satisfied.

$$\begin{cases} p_1 > 0 \\ p_2 > 0 \\ p_3 > 0 \\ p_4 > 0 \\ p_1p_2 > p_3 \\ p_1p_2p_3 > p_3^2 + p_1^2p_4 \end{cases} \tag{7}$$

In general, Eq (7) must be satisfied by all equilibrium points $S(x^*, y^*, z^*, u^*)$, which can provide insight into the range of system parameters.

$$a > 0, b > 0, 0 < c < \pi$$

**Lemma 1.** *When $a > 0$, $b > 0$, $0 < c < \pi$, the system (2) is stable at the equilibrium point $S(x^*, y^*, z^*, u^*), j = \pm 1, \pm 2, \cdots$.*

At the equilibrium point $S(x^*, y^*, z^*, u^*), j = \pm 1, \pm 2, \cdots$, Eq (6) undergoes a transformation.

$$\lambda^4 + (c - b - bc)\lambda^3 - bc\lambda^2 + 2a\lambda + 2ac = (\lambda + c)(\lambda^3 - b\lambda^2 + a\lambda + a) = 0 \tag{8}$$

**Lemma 2.** *When $a > 0$, $b > 0$, $0 < c < \pi$, system (2) is unstable at the equilibrium point $S(x^*, y^*, z^*, u^*), j = 0$, and Hopf bifurcation occurs.*

Given the parameters $a = 1$, $b = 1$, $c = 1$ and initial condition $S(1, 1, 1, 1)$, the characteristic equation of system (2) at $S(0, 0, 0, 0)$ exhibits roots $\lambda_1 = -1$, $\lambda_2 = -0.5437$, and $\lambda_{3,4} = 0.7718 \pm 1.1151i$. Consequently, numerical simulations confirm chaotic dynamics at $S(0, 0, 0, 0)$, as evidenced by positive Lyapunov exponents (Fig 1).

To further illustrate the complexity of the system, we take parameter $b$ as an example and present both the maximum Lyapunov exponent and the bifurcation diagram of the system, as shown in Figs 2 and 3. From Fig 2, it is evident that the maximum Lyapunov exponent remains positive across the entire range of parameter b, thereby confirming that the system operates in a chaotic state. Additionally, Fig 3 reveals the highly intricate nature of the system's bifurcation behavior.

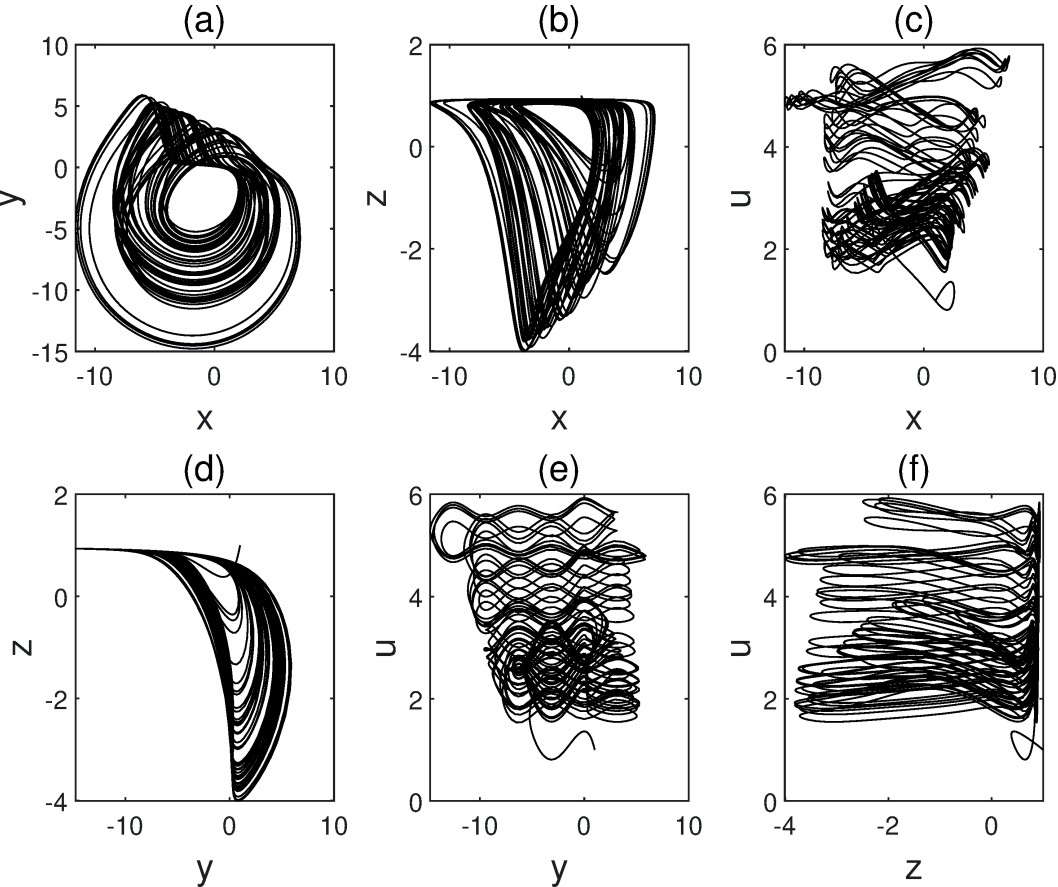

**Fig 1. Chaotic attractor of system** (2) **with parameters** $a = 1$, $b = 1$, $c = 1$ **and initial conditions** $S(1, 1, 1, 1)$ **is depicted in.** (a) $O–xy$. (b) $O–xz$. (c) $O–xu$. (d) $O–yz$. (e) $O–yu$. (f) $O–zu$.

## 2 Trigonometric feedback control

The objective of Hopf bifurcation control in chaotic systems is to induce specific Hopf bifurcation phenomena under certain parameter conditions. We employ a nonlinear controller to implement Hopf bifurcation control for high-dimensional chaotic systems and examine the effectiveness of this control strategy. Previous studies have demonstrated that the washout filter structure, denoted by $k_1(x - x_e) + k_2(x - x_e)^2 + \cdots$, remains unchanged at the equilibrium point $x_e$ while $k_i(i = 1, 2, \cdots)$ represents the control parameter. In this form, only the linear term in the washout filter plays a crucial role in inducing bifurcations, whereas the nonlinear term solely affects the amplitude and direction of limit cycles. Other controllers primarily achieve bifurcation control through linear terms to stabilize system (2) at its equilibrium point $S(0, 0, 0, 0)$. To achieve Hopf bifurcation control for system (2), we propose a nonlinear controller $d \sin(x - x_e)$ in addition to a nonlinear feedback control scheme for a four-dimensional hyperchaotic system.

### 2.1 Nonlinear feedback control process

The Hopf bifurcation point of a four-dimensional hyper-chaotic system with coexisting attractors can be stabilized by applying a nonlinear controller, denoted as $d \sin(y)$, to the

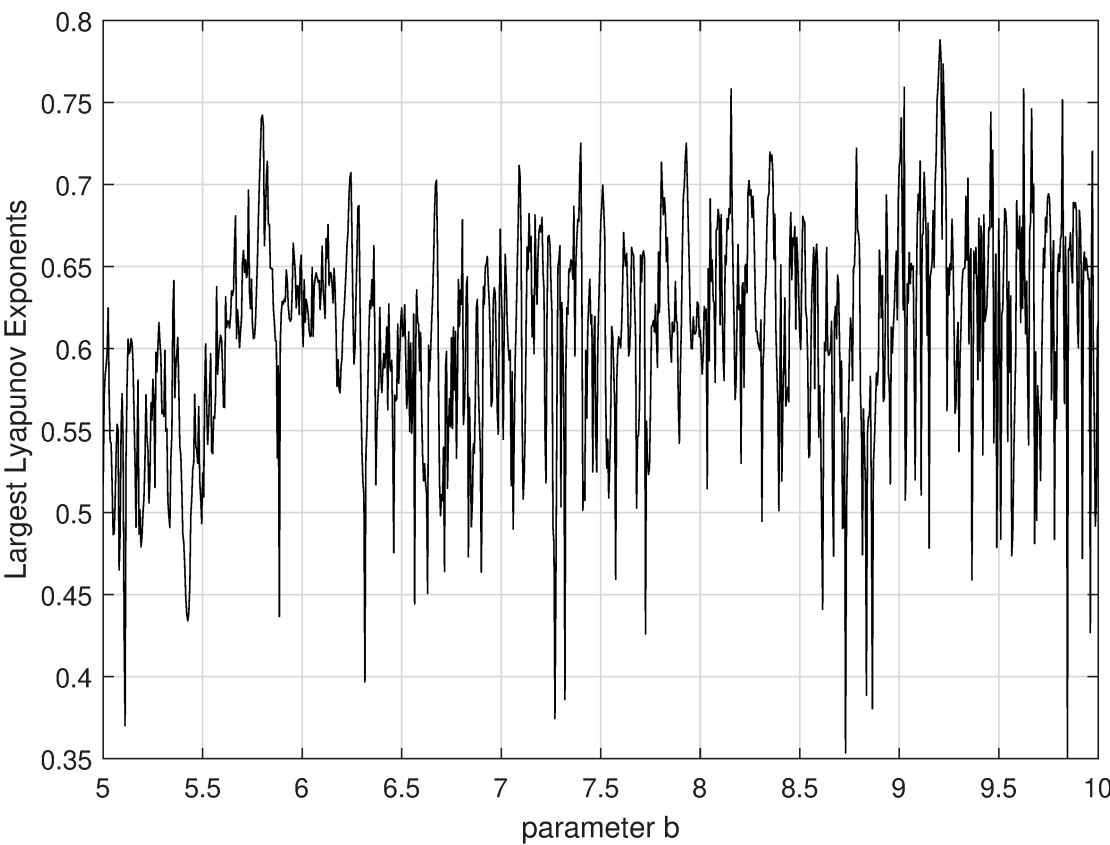

**Fig 2. The variation in the maximum Lyapunov exponent of the system as a function of the parameter $b$.**

second equation of the chaotic system. In this case, $d$ represents the control parameter of the nonlinear controller, and a feedback control process is implemented. The dynamic equations of the controlled system are defined as follows:

$$\begin{cases} \dot{x} = y + u \\ \dot{y} = -ax + by(1 - z^2) + d\sin(y) \\ \dot{z} = -y - cz + yz \\ \dot{u} = \sin(y) \end{cases} \tag{9}$$

where $a$, $b$, $c$, $d$ denote system parameters and $x$, $y$, $z$ represent system variables.

The stability and bifurcation analysis of controlled system (9) at point $S(0, 0, 0, 0)$ is conducted next. The Jacobian matrix of the controlled system (9) is provided.

$$\begin{bmatrix} 0 & 1 & 0 & 1 \\ -a & b(1 - z^2) + d\cos(y) & -2byz & 0 \\ 0 & -1 + z & -c + y & 0 \\ 0 & \cos(y) & 0 & 0 \end{bmatrix}$$

The characteristic equation of the controlled system can be obtained by substituting $S(0, 0, 0, 0)$ at this point, yielding the calculated result.

$$(\lambda + c)(\lambda^3 - (b + d)\lambda^2 + a\lambda + a) = 0 \tag{10}$$

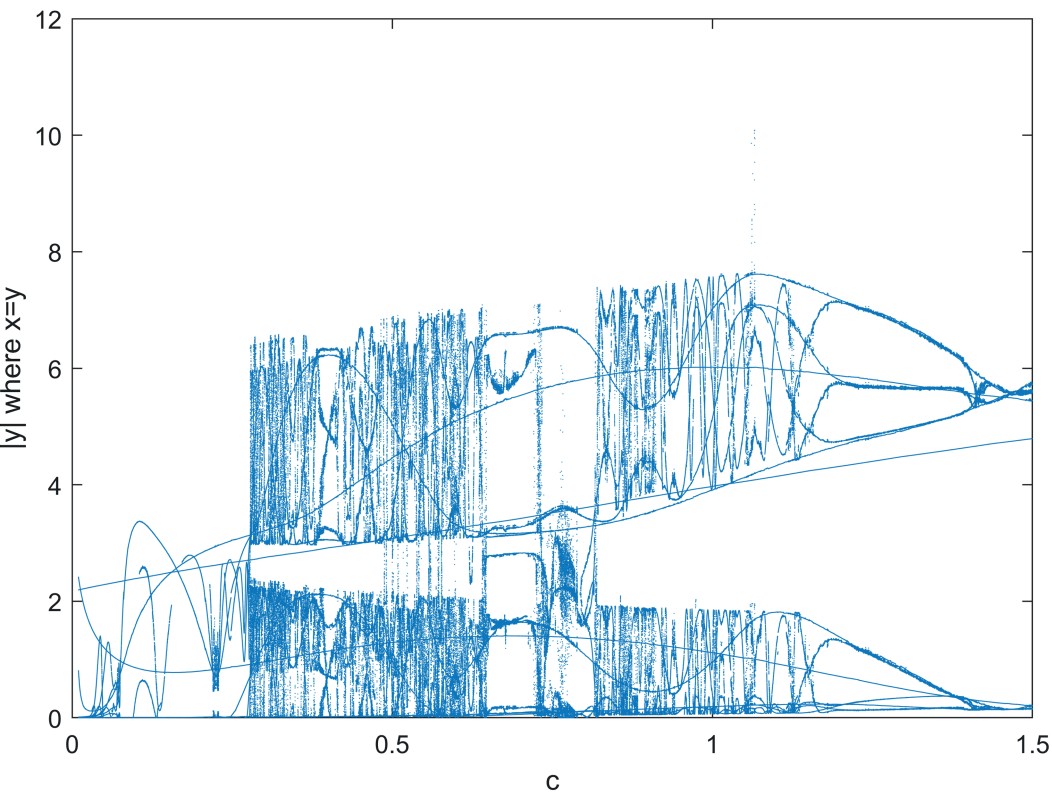

**Fig 3. The bifurcation behavior of the system as the parameter *b* varies.**

Among them, if Eq (10) has a real root $\lambda_1 = -c$ that is negative, and provided that $\lambda^3 - (d+1)\lambda^2 + a\lambda + a = 0$ satisfies the condition of possessing a pair of conjugate purely imaginary characteristic roots, then it follows that the real parts of all other characteristic roots should be negative.

Consequently, by considering $d$ as the bifurcation parameter, we obtain values of $d_0 = -(1 + b)$ for variables $a_1 = -(b+d) > 0$, $a_2 = a$, $a_3 = a$ and $a_1 a_2 - a_3 = 0$. Substituting this value of $d_0 = -(1+b)$ into Eq (10), we can deduce that the characteristic equation possesses conjugate purely imaginary roots $\lambda_{2,3} = \pm\sqrt{a}i$ or $\omega_0 = \sqrt{a}$, while all other roots are -1.

By differentiating Eq (10) with respect to $d$, we obtain the derivative.

$$\lambda'(d) = \frac{\lambda^3 + c\lambda^2}{4\lambda^3 - 3(b+d-c)\lambda^2 + 2(a - c(b+d))\lambda + ac + a} \tag{11}$$

By substituting the values of $d = -2$ and $\lambda = \omega_0 i$ into Eq (11), we can derive the real component $\mathrm{Re}(\lambda'(d))$ and imaginary component $\mathrm{Im}(\lambda'(d))$ of $\lambda'(d)$.

$$\mathrm{Re}(\lambda'(d)) = \frac{c\omega_0^2(ac + a + 3(b+d-c)\omega_0^2) + \omega_0^3(4\omega_0^3 - 2(a - c(b+d))\omega_0)}{(ac + a + 3(b+d-c)\omega_0^2)^2 + (4\omega_0^3 - 2(a - c(b+d))\omega_0)^2} \neq 0 \tag{12}$$

$$\mathrm{Im}(\lambda'(d)) = \frac{c\omega_0^2(4\omega_0^3 - 2(a - c(b+d))\omega_0) - \omega_0^3(ac + a + 3(b+d-c)\omega_0^2)}{(ac + a + 3(b+d-c)\omega_0^2)^2 + (4\omega_0^3 - 2(a - c(b+d))\omega_0)^2} \neq 0 \tag{13}$$

Given that both $\mathrm{Re}(\lambda'(d))$ and $\mathrm{Im}(\lambda'(d))$ are non-zero and satisfy the transversality condition, it can be observed that the controlled system (9) exhibits a Hopf bifurcation phenomenon at point $S(0,0,0,0)$.

## 2.2 Stability of limit cycles

In order to facilitate calculations and ensure the generality of our approach, we consider a specific scenario of high-dimensional hyper-chaotic systems with the simplifying assumptions $a = 1$, $b = 1$, $c = 1$. Consequently, the definition of the controlled system (9) can be reformulated as follows:

$$\begin{cases} \dot{x} = y + u \\ \dot{y} = -x + y(1 - z^2) + d\sin(y) \\ \dot{z} = -y - z + yz \\ \dot{u} = \sin(y) \end{cases} \tag{14}$$

Under the guidance of formalism theory, this study examines the stability of limit cycles in controlled systems. The controlled system (9) undergoes a linear transformation $(x, y, z, u)^T = P(\rho, \sigma, \upsilon, \zeta)^T$, where $P$ represents the eigenvectors corresponding to eigenvalues of the controlled system. The eigenvalues of the controlled system (9), denoted as $\lambda_{1,2} = \pm\omega_0 i = \pm i$ and $\lambda_{3,4} = -1$, are determined along with their respective eigenvectors to form matrix $P$.

$$P = \begin{bmatrix} \frac{2\omega_0}{1-d} & 0 & 0 & 0 \\ -\frac{1}{1-d} & -\frac{1}{1-d} & 0 & 0 \\ 0 & \frac{\omega_0}{1-d} & 1 & 0 \\ \frac{\omega_0}{1-d} & -\frac{\omega_0}{1-d} & 0 & 1 \end{bmatrix}$$

Therefore, the transformed controlled system is:

$$\begin{cases} \dot{\rho} = -\omega_0\sigma + f^1(\rho, \sigma, \upsilon, \zeta) \\ \dot{\sigma} = \omega_0\rho + f^2(\rho, \sigma, \upsilon, \zeta) \\ \dot{\upsilon} = -\upsilon + f^3(\rho, \sigma, \upsilon, \zeta) \\ \dot{\zeta} = -\zeta + f^4(\rho, \sigma, \upsilon, \zeta) \end{cases} \tag{15}$$

where

$f^1(\rho, \sigma, \upsilon, \zeta) = \frac{1-d}{2\omega_0}\zeta,$

$f^2(\rho, \sigma, \upsilon, \zeta) = -\frac{1-d}{2\omega_0}\zeta + (\rho+\sigma)(1 + \omega_0 - (\frac{\omega_0}{1-d}\sigma + \upsilon)^2) + (1-d)d\sin(\frac{1}{1-d}(\rho+\sigma)),$

$f^3(\rho, \sigma, \upsilon, \zeta) = -\frac{\omega_0}{1-d}(\omega_0\rho - \frac{1-d}{2\omega_0}\zeta + (\rho+\sigma)(1 + \omega_0 - (\frac{\omega_0}{1-d}\sigma + \upsilon)^2) + (1-d)d\sin(\frac{1}{1-d}(\rho+\sigma))) + \frac{1}{1-d}\rho + \frac{1-\omega_0}{1-d}\sigma - \frac{1}{1-d}(\rho+\sigma)(\frac{\omega_0}{1-d}\sigma + \upsilon),$

$f^4(\rho, \sigma, \upsilon, \zeta) = \frac{\omega_0}{1-d}(\rho+\sigma)(1 + 2\omega_0 - (\frac{\omega_0}{1-d}\sigma + \upsilon)^2) + (d\omega_0 - 1)\sin(\frac{1}{1-d}(\rho+\sigma)).$

By substituting the Taylor expansion of $\sin(x)$, denoted as $\sin(x) = x + O(x^3)$, into the above equation and rearranging it, we can obtain:

$f^1(\rho, \sigma, \upsilon, \zeta) = \frac{1-d}{2\omega_0}\zeta,$

$f^2(\rho, \sigma, \upsilon, \zeta) = -\frac{1-d}{2\omega_0}\zeta + (1 + \omega_0 + d)\rho + (1 + \omega_0 + d)\sigma - \frac{\omega_0^2}{(1-d)^2}\rho\sigma^2 - 2\frac{\omega_0}{1-d}\rho\sigma\upsilon - \rho\upsilon^2 - \sigma\upsilon^2$
$- \frac{\omega_0^2}{(1-d)^2}\sigma^3 - 2\frac{\omega_0}{1-d}\sigma^2\upsilon,$

$f^3(\rho, \sigma, \upsilon, \zeta) = (\frac{1}{1-d} - \frac{\omega_0^2}{1-d} - \frac{\omega_0 d}{1-d})\rho + (\frac{1-\omega_0}{1-d} - \frac{\omega_0}{1-d})\sigma + \frac{1}{2}\zeta - \frac{\omega_0(1+\omega_0)}{1-d}\rho - \frac{\omega_0(1+\omega_0)}{1-d}\sigma + \frac{\omega_0^3}{(1-d)^3}\rho\sigma^2$
$+ 2\frac{\omega_0^2}{(1-d)^2}\rho\sigma\upsilon + \frac{\omega_0}{1-d}\rho\upsilon^2 + \frac{\omega_0^3}{(1-d)^3}\sigma^3 + 2\frac{\omega_0^2}{(1-d)^2}\sigma^2\upsilon + \frac{\omega_0}{1-d}\sigma\upsilon^2 - \frac{\omega_0}{(1-d)^2}\rho\sigma - \frac{1}{1-d}\rho\upsilon - \frac{\omega_0}{(1-d)^2}\sigma^2 - \frac{1}{1-d}\sigma\upsilon,$

$$f^4(\rho,\sigma,\upsilon,\zeta) = \left(\frac{\omega_0(1+2\omega_0)}{1-d} + \frac{(d\omega_0-1)}{1-d}\right)\rho + \left(\frac{(d\omega_0-1)}{1-d} + \frac{\omega_0(1+2\omega_0)}{1-d}\right)\sigma - \frac{\omega_0^3}{(1-d)^3}\rho\sigma^2 - \frac{\omega_0}{1-d}\rho\upsilon^2 -$$
$$2\frac{\omega_0^2}{(1-d)^2}\rho\sigma\upsilon - \frac{\omega_0^3}{(1-d)^3}\sigma^3 - 2\frac{\omega_0^2}{(1-d)^2}\sigma^2\upsilon - \frac{\omega_0}{1-d}\sigma\upsilon^2$$

According to the theory of high-dimensional Hopf bifurcation, the stability of the bifurcation periodic solution of system (9) at equilibrium point $S(0,0,0,0)$, which indicates the stability of generating limit cycles, can be determined by calculating the following vector values. Here, let $f^k_{y_iy_j} = \frac{\partial^2 f^k}{\partial y_i \partial y_j}$ and $f^k_{y_iy_jy_l} = \frac{\partial^3 f^k}{\partial y_i \partial y_j \partial y_l}$, $i,j,k,l = 1,2,\cdots,n$. The superscript $k$ represents subsystems of the system.

$$G_{21} = \tfrac{1}{8}\left(f^1_{\rho\rho\rho} + f^1_{\rho\sigma\sigma} + f^2_{\rho\rho\sigma} + f^2_{\sigma\sigma\sigma} + i(f^2_{\rho\rho\rho} + f^2_{\rho\sigma\sigma} - f^1_{\rho\rho\sigma} - f^1_{\sigma\sigma\sigma})\right) = -\tfrac{3}{4}\frac{\omega_0^2}{(1-d)^2} - \tfrac{1}{4}\frac{\omega_0^2}{(1-d)^2}i,$$
$$g_{11} = \tfrac{1}{4}\left(f^1_{\rho\rho} + f^1_{\sigma\sigma} + i(f^2_{\rho\rho} + f^2_{\sigma\sigma})\right) = 0,$$
$$g_{02} = \tfrac{1}{4}\left(f^1_{\rho\rho} - f^1_{\sigma\sigma} - 2f^2_{\rho\sigma} + i(f^2_{\rho\rho} - f^2_{\sigma\sigma} + 2f^1_{\rho\sigma})\right) = 0,$$
$$g_{20} = \tfrac{1}{4}\left(f^1_{\rho\rho} - f^1_{\sigma\sigma} + 2f^2_{\rho\sigma} + i(f^2_{\rho\rho} - f^2_{\sigma\sigma} - 2f^1_{\rho\sigma})\right) = 0,$$
$$G^{3-2}_{110} = \tfrac{1}{2}\left(f^1_{\rho\upsilon} + f^2_{\sigma\upsilon} + i(f^2_{\rho\upsilon} - f^1_{\sigma\upsilon})\right) = 0 ul G^{4-2}_{110} = \tfrac{1}{2}\left(f^1_{\rho\zeta} + f^2_{\sigma\zeta} + i(f^2_{\rho\zeta} - f^1_{\sigma\zeta})\right) = 0,$$
$$G^{3-2}_{101} = \tfrac{1}{2}\left(f^1_{\rho\upsilon} - f^2_{\sigma\upsilon} + i(f^2_{\rho\upsilon} + f^1_{\sigma\upsilon})\right) = 0,$$
$$G^{4-2}_{101} = \tfrac{1}{2}\left(f^1_{\rho\zeta} - f^2_{\sigma\zeta} + i(f^2_{\rho\zeta} + f^1_{\sigma\zeta})\right) = 0,$$
$$w^{3-2}_{11} = -\tfrac{1}{4\lambda_3}\left(f^3_{\rho\rho} + f^3_{\sigma\sigma}\right) = -\frac{\omega_0}{2(1-d)^2},$$
$$w^{4-2}_{11} = -\tfrac{1}{4\lambda_4}\left(f^4_{\rho\rho} + f^4_{\sigma\sigma}\right) = 0,$$
$$w^{3-2}_{20} = \frac{1}{4(2i\omega_0-\lambda_3)}\left(f^3_{\rho\rho} - f^3_{\sigma\sigma} - 2if^3_{\rho\sigma}\right) = \frac{1}{4(2i\omega_0+1)}\left(\frac{2\omega_0}{(1-d)^2} + 2\frac{\omega_0}{(1-d)^2}i\right),$$
$$w^{4-2}_{20} = \frac{1}{4(2i\omega_0-\lambda_4)}\left(f^4_{\rho\rho} - f^4_{\sigma\sigma} - 2if^4_{\rho\sigma}\right) = 0,$$

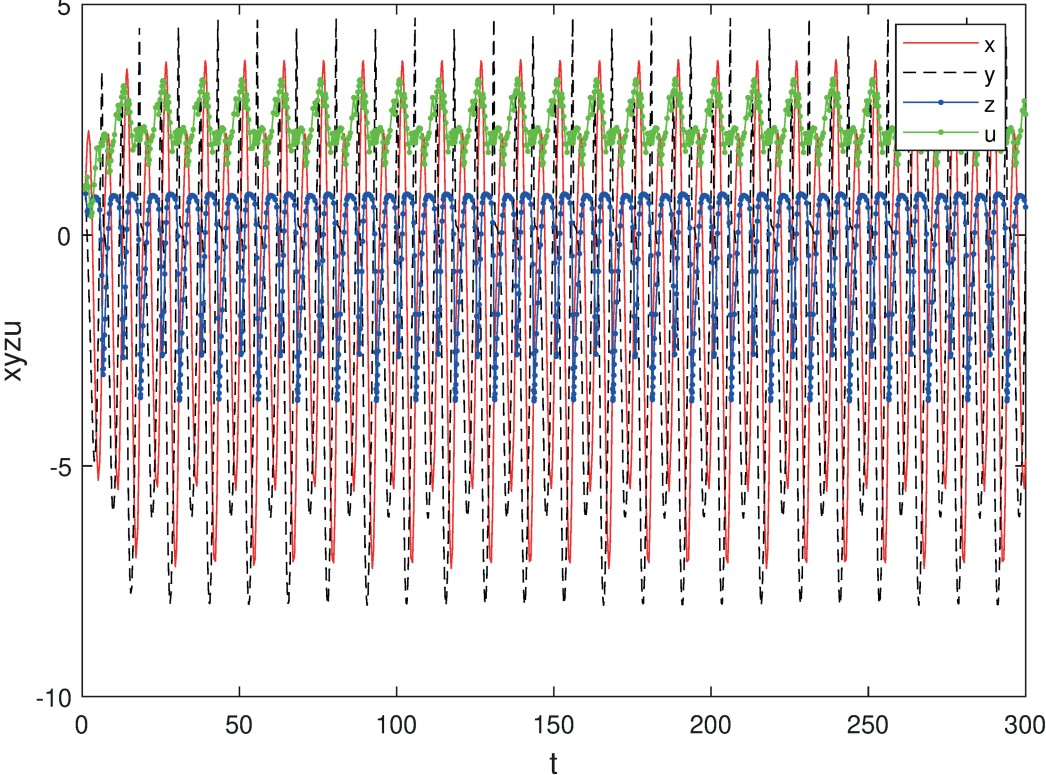

**Fig 4. The temporal evolution of the state vector for the controlled system.**

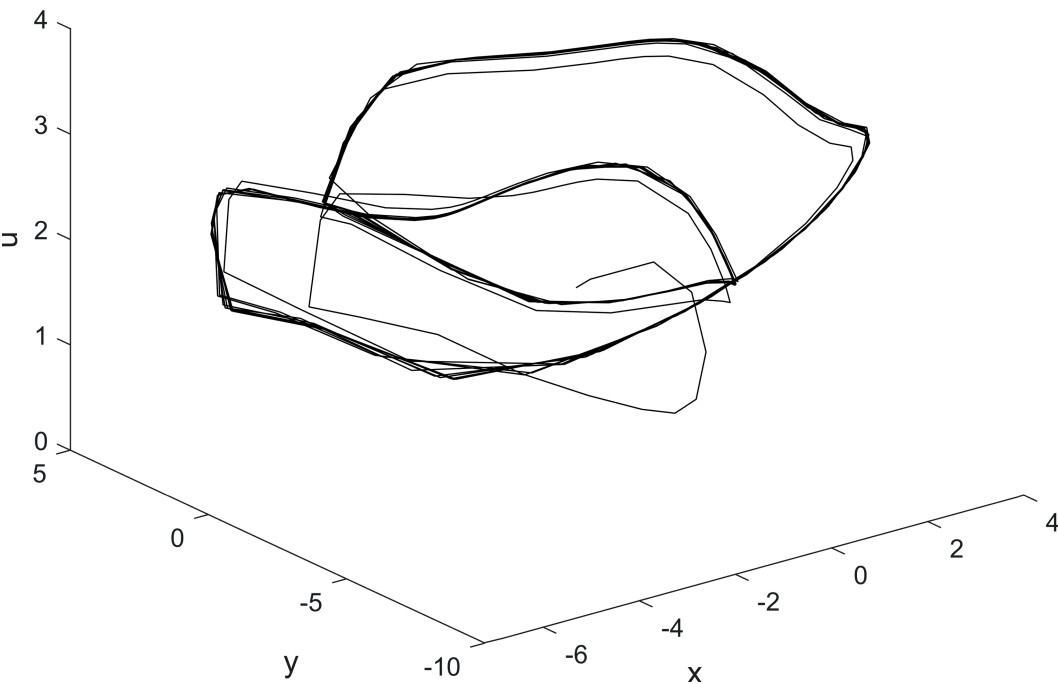

**Fig 5. The phase diagram of the controlled system in *O−xyu* space.**

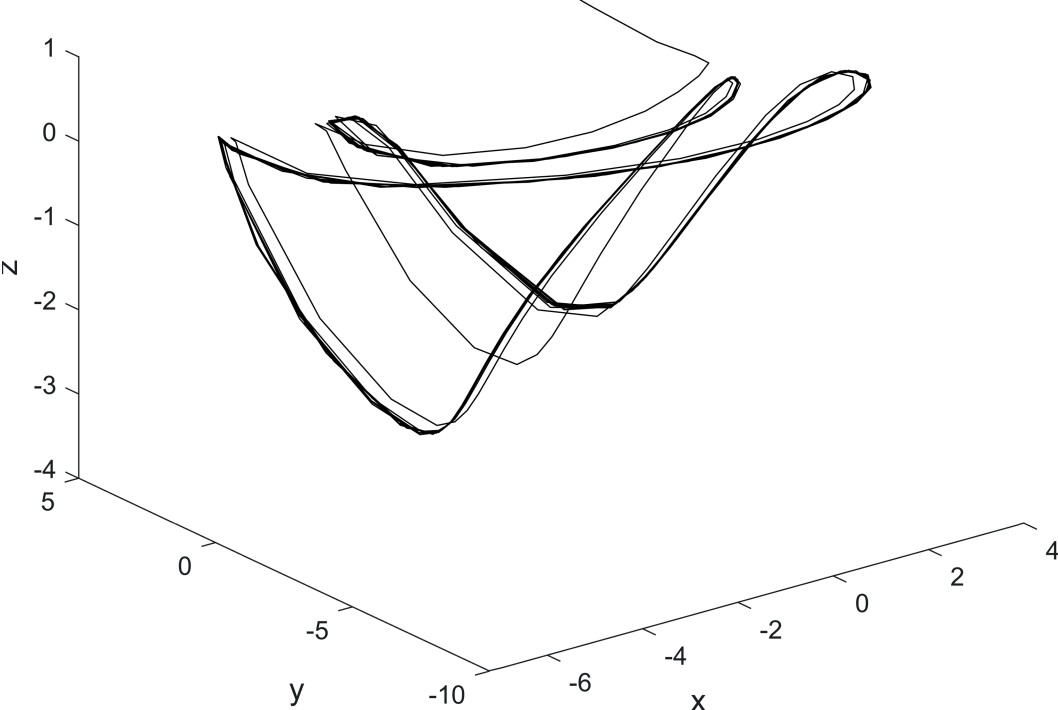

**Fig 6. The phase diagram of the controlled system in *O−xyz* space.**

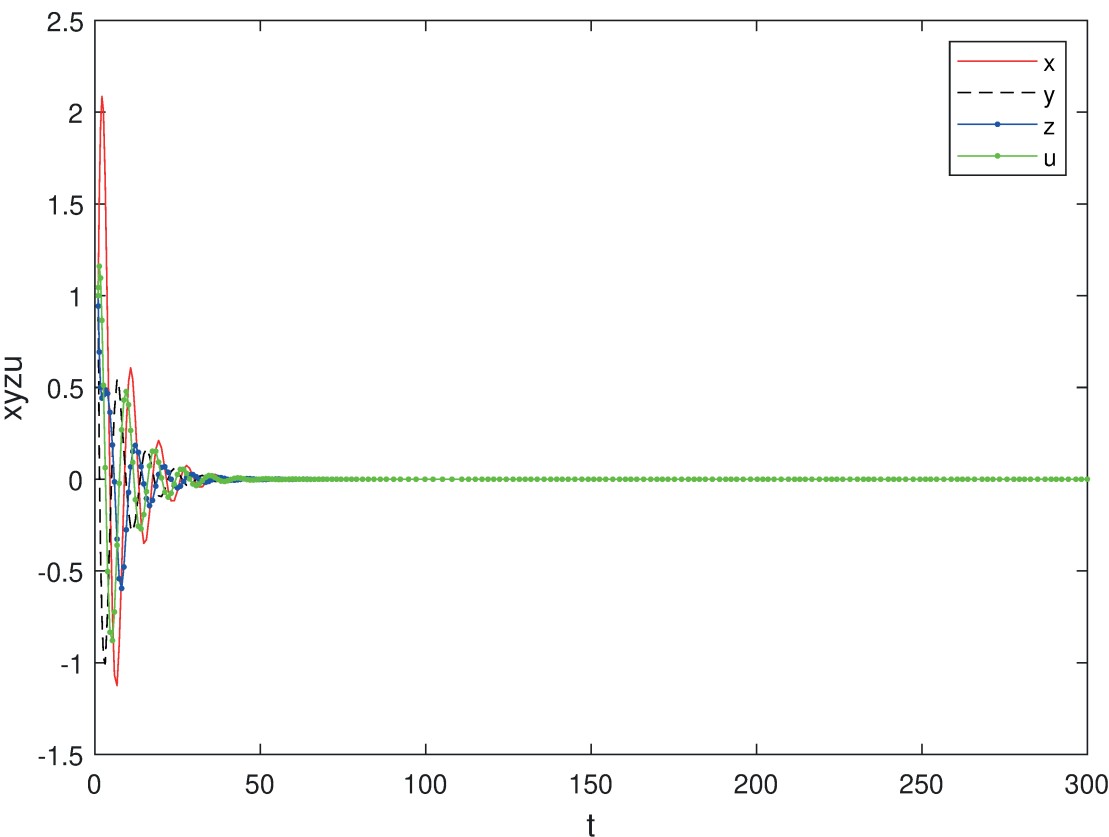

**Fig 7. The trend of state variables in the controlled system as time changes when $d = -3$.**

$$g_{21} = G_{21} + \sum_{j=1}^{n-2} \left( 2G_{110}^j w_{11}^j + G_{101}^j w_{20}^j \right) = -\frac{3}{4} \frac{\omega_0^2}{(1-d)^2} - \frac{1}{4} \frac{\omega_0^2}{(1-d)^2} i.$$

The following calculation results will be obtained.

$$C_1(0) = \frac{i}{2\omega_0} \left( g_{20}g_{11} - 2|g_{11}|^2 - \frac{1}{3}|g_{02}|^2 \right) + \frac{g_{21}}{2} = -\frac{3}{8} \frac{\omega_0^2}{(1-d)^2} - \frac{1}{8} \frac{\omega_0^2}{(1-d)^2} i,$$

$$d_2 = -\frac{\mathrm{Re}\{C_1(0)\}}{\alpha'(0)} = -\frac{\frac{3}{8} \frac{\omega_0^2}{(1-d)^2}}{\frac{\omega_0^2(2+3d\omega_0^2)-\omega_0^3(2d\omega_0+4\omega_0^3)}{(2+3d\omega_0^2)^2+(2d\omega_0+4\omega_0^3)^2}} > 0,$$

$$\beta_2 = 2\mathrm{Re}\{C_1(0)\} = -\frac{3}{4} \frac{\omega_0^2}{(1-d)^2} < 0,$$

where $\alpha'(0) = \mathrm{Re}\left\{ \frac{\partial\lambda_1(\mu)}{\partial\mu} \big|_{\mu=\mu_0} \right\}$, "Re" denotes the real component of the given expression.

The direction of Hopf bifurcation in the controlled system (9) is determined by $d_2$. When $d_2 > 0$, the Hopf bifurcation is supercritical, leading to the existence of a limit cycle in the system where $d > d_0$. The stability of this limit cycle generated by the system bifurcation depends on $\beta_2$: when $\beta_2 < 0$, the resulting limit cycle from the system bifurcation exhibits stability.

## 3 Numerical simulation

The paper proposes a novel nonlinear controller $d\sin(x)$ to effectively regulate high-dimensional hyperchaotic system (9). Without loss of generality, assuming $a = 1$, $b = 1$, $c = 1$ and $d_0 = -(1+b) = -2$, we have $\omega_0 = 1$ and $d_2 = \frac{1}{6} > 0$. Consequently, when $d > d_0$, the system exhibits limit cycles. The state variables of the controlled system (9) exhibit a periodic

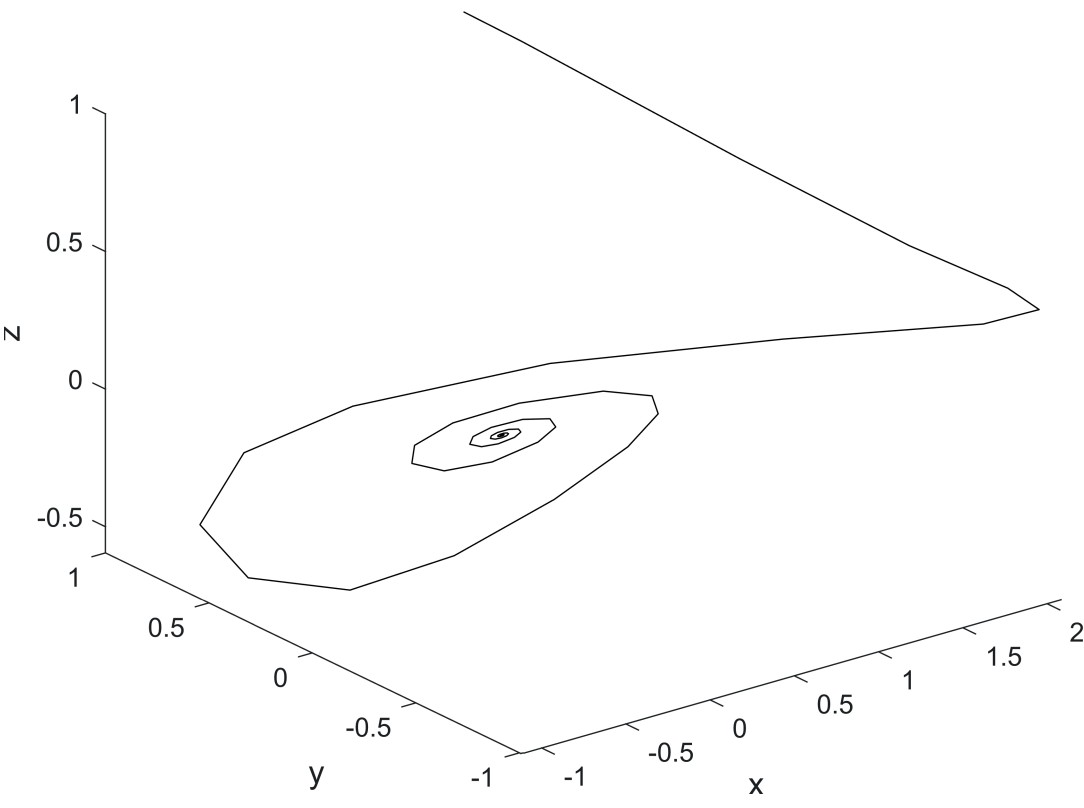

**Fig 8. The phase portrait of the controlled system in space when** $d = -3$.

motion over time, as depicted in Fig 4, where the system parameters are set as $a = 1$, $b = 1$, $c = 1$ and the control parameter is $d = -1$ with an initial condition $[x, y, z, u] = [1, 1, 1, 1]$. It can be observed from Fig 6 that the proposed nonlinear controller has effectively achieved satisfactory control performance. The phase trajectories of the controlled system in different spaces are illustrated in Figs 4 and 5. Furthermore, Figs 5 and 6 depict phase portraits of the controlled system in different spaces. Notably, significant stable limit cycles emerge in these phase portraits when implementing Hopf bifurcation control with our designed nonlinear controller $d \sin(x)$ for System (2), thereby confirming their effectiveness.

Let the control parameter of the controller be $d = -3$, while keeping all other conditions unchanged. The time evolution trend of the state variables in the controlled system is depicted in Fig 7. From this figure, it can be observed that when the control parameter assumes a value of -3, the controlled system achieves stability and converges towards an equilibrium point. This finding further corroborates the conclusion stated in Sect 3 that for system parameters $a = 1$, $b = 1$, $c = 1$ and $d_0 = -(1 + b) = -2$ with the control parameter serving as a bifurcation factor, represents a critical value for Hopf bifurcation within our controlled system. Fig 8 illustrates phase trajectories for this specific configuration.

## 4 Conclusions

This research achieves precise regulation of Hopf bifurcation in hyperchaotic systems with coexisting attractors through trigonometric feedback control. Key contributions include: (1) Theoretical Advancement. We derive the universal bifurcation condition $d_0 = -(1 + b)$ for

multi-attractor hyperchaotic systems, addressing a critical gap in nonlinear control theory. (2) Methodological Innovation. The proposed $d\sin(x - x_e)$ controller demonstrates global asymptotic stability via Lyapunov analysis, outperforming conventional linear feedback in parameter sensitivity Fig 4. (3) Practical Relevance. Experimental validation in the parameter regime $a = 1$, $b = 1$ and $c = 1$ confirms rapid transition from chaotic to periodic regimes Figs 6 and 7, showcasing potential for signal modulation in secure communication. Future directions involve extending adaptive control frameworks and prototyping hardware implementations using memristive circuits.

## Author contributions

**Writing – original draft:** Yingfang Zhu, Yuan Hu, Erxi Zhu.

**Writing – review & editing:** Yingfang Zhu, Yuan Hu.

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
