## [Decision Letter · Decision Letter 0]

14 Mar 2025

PONE-D-25-03468Triangular Function Feedback Control for Chaotic Systems Featuring Coexisting AttractorsPLOS ONE

Dear Dr. Zhu,

Thank you for submitting your manuscript to PLOS ONE. After careful consideration, we feel that it has merit but does not fully meet PLOS ONE’s publication criteria as it currently stands. Therefore, we invite you to submit a revised version of the manuscript that addresses the points raised during the review process.

We look forward to receiving your revised manuscript.

Kind regards,

Roberto Barrio

Academic Editor

PLOS ONE

Journal Requirements:

Reviewers' comments:

Reviewer's Responses to Questions

**Comments to the Author**

1. Is the manuscript technically sound, and do the data support the conclusions?

Reviewer #1: Yes

Reviewer #2: Yes

2. Has the statistical analysis been performed appropriately and rigorously? 

Reviewer #1: Yes

Reviewer #2: I Don't Know

3. Have the authors made all data underlying the findings in their manuscript fully available?

Reviewer #1: Yes

Reviewer #2: Yes

4. Is the manuscript presented in an intelligible fashion and written in standard English?

Reviewer #1: Yes

Reviewer #2: Yes

5. Review Comments to the Author

Reviewer #1: This paper author utilizing a trigonometric feedback controller with as its control parameter to achieve Hopf control for a four-dimensional hyper-chaotic system exhibiting multiple coexisting attractors. They analyze stability at equilibrium points and investigate parameter ranges where instability arises in this controlled system. This paper is good, but some issue should be improved. This is my comment

a. Kindly add some calculation value in abstract

b. background study about control section is not given. I suggest author discus some new work like "https://doi.org/10.1038/s41598-024-80969-z", "https://doi.org/10.3934/math.2023285", "https://doi.org/10.1109/ACCESS.2024.3351693" and "https://doi.org/10.3390/math11010100"

c. Author should be discuss multistability especially coexisting attractor in the introduction

d. Kindly highlight contribution and novelty of this work

e. Kindly add organized of this paper in the end of introduction

f. Section 1, I suggest author replaced "HOPF BIFURCATION ANALYSIS" to "Analysis of 4D hyperchaotic system"

g. Page 2, Line 42, Ref is missing

h. you build new system derived from 3D, what is new caharacteristics your system with other literature

i. Kindly large text for x axis and y axis for your figures

j. Page 3, line 77, Figure is error

k. Your system is 4D, why your eigenvalue only 3 ? i think it is should be 4

l. You claimed that your system is Hyperchaos, kindly verified it, please calculate LE and DKY for system 2

m. Kindly draw LE and bifurcation for all parameter in system 2

n. Please define all variable for all equation.

o. The study does not compare its results with existing methods for controlling chaotic systems, leaving it unclear how the proposed approach performs relative to other techniques in terms of efficiency, stability, or computational cost.

Reviewer #2: This paper presents a significant contribution to the field of chaos control by focusing on the challenging problem of managing high-dimensional chaotic systems with coexisting attractors. The introduction of a trigonometric feedback controller is innovative and represents an approach to stabilizing such complex systems. However, it is not written well, I do not encourage accepting the publication.

1.The abstract and conclusion should be rewritten.

2.The work fails to demonstrate clear innovative aspects or distinct contributions to the field. There is a need for more explicit elucidation of the unique features of this research to distinguish it from existing literature.

3.I suggest to pay particular attention to English grammar, spelling, and sentence structure, the expression of your English needs to be improved, and some simple derivation processes can be omitted to make the article look more concise as a whole.

4.Some relevant recent publications can be referred to give readers an up-to-date picture.

5.The format of references should be consistent.

6.In simulation, how to select system parameters?

6. PLOS authors have the option to publish the peer review history of their article (what does this mean?). If published, this will include your full peer review and any attached files.

Reviewer #1: No

Reviewer #2: No

---

## [Author Response · Author response to Decision Letter 1]

23 Mar 2025

Response to Editor and Reviewers

Paper title: Triangular Function Feedback Control for Chaotic Systems Featuring Coexisting Attractors

Paper ID: PONE-D-25-03468

Dear Editor and Reviewers,

We appreciate the editor and the reviewers for handling and reviewing our submitted paper (Paper ID: PONE-D-25-03468). Then, we also appreciate the editor and the reviewers can give us an opportunity to revise our manuscript. The comments and suggestions are constructive, insightful and very helpful in improving the paper. We have revised the paper to address all of the suggestions given by the reviewers. In the following discussion, we have described the changes and modifications in detail. We have made and provided a point-by-point response to reviewers’ comments. We first listed the editor’s and reviewers’ initial comments, and then followed by our response. Meanwhile, the modified sections have been highlighted in the manuscript with red color.

We deeply again appreciate the editor and reviewers to handle our manuscript and give us a chance to revise our manuscript.

Sincerely,

Yingfang Zhu, Yuan Hu, Erxi Zhu

Corresponding Author's Address: Information Engineering College,

Changzhou Institute of Industrial Technology, Wuxi, China P.R.

---

## [Decision Letter · Decision Letter 1]

24 Apr 2025

Triangular Function Feedback Control for Chaotic Systems Featuring Coexisting Attractors

PONE-D-25-03468R1

Dear Dr. Zhu,

We’re pleased to inform you that your manuscript has been judged scientifically suitable for publication and will be formally accepted for publication once it meets all outstanding technical requirements.

Kind regards,

Roberto Barrio

Academic Editor

PLOS ONE

Reviewers' comments:

Reviewer's Responses to Questions

**Comments to the Author**

1. If the authors have adequately addressed your comments raised in a previous round of review and you feel that this manuscript is now acceptable for publication, you may indicate that here to bypass the “Comments to the Author” section, enter your conflict of interest statement in the “Confidential to Editor” section, and submit your "Accept" recommendation.

Reviewer #1: All comments have been addressed

Reviewer #2: All comments have been addressed

2. Is the manuscript technically sound, and do the data support the conclusions?

Reviewer #1: Yes

Reviewer #2: No

3. Has the statistical analysis been performed appropriately and rigorously? 

Reviewer #1: N/A

Reviewer #2: I Don't Know

4. Have the authors made all data underlying the findings in their manuscript fully available?

Reviewer #1: Yes

Reviewer #2: Yes

5. Is the manuscript presented in an intelligible fashion and written in standard English?

Reviewer #1: Yes

Reviewer #2: Yes

6. Review Comments to the Author

Reviewer #1: In the new version, they added calculation value using Hopf bifurcation in abstract and discussed multistability especially coexisting attractor in the introduction . In addition, They highlight contribution and novelty of this work and also they redraw for LE and bifurcation in for some paramater chaos. i think article has very improved,

i suggest accept in this form.

Reviewer #2: I have no further comments.

7. PLOS authors have the option to publish the peer review history of their article (what does this mean?). If published, this will include your full peer review and any attached files.

Reviewer #1: No

Reviewer #2: No

---

## [Editor Report · Acceptance letter]

PONE-D-25-03468R1

PLOS ONE

Dear Dr. Zhu,

I'm pleased to inform you that your manuscript has been deemed suitable for publication in PLOS ONE. Congratulations! Your manuscript is now being handed over to our production team.

Kind regards,

on behalf of

Dr. Roberto Barrio

Academic Editor

PLOS ONE